# Gene Expression Dysregulation in Whole Blood of Patients with *Clostridioides difficile* Infection

**DOI:** 10.3390/ijms252312653

**Published:** 2024-11-25

**Authors:** Maria Tsakiroglou, Anthony Evans, Alejandra Doce-Carracedo, Margaret Little, Rachel Hornby, Paul Roberts, Eunice Zhang, Fabio Miyajima, Munir Pirmohamed

**Affiliations:** 1Department of Pharmacology and Therapeutics, Institute of Systems Molecular and Integrative Biology, University of Liverpool, Liverpool L69 3GL, UK; adoce@liverpool.ac.uk (A.D.-C.); margaret.little@liverpool.ac.uk (M.L.); rachael.hornby@liverpoolft.nhs.uk (R.H.); paul.roberts@wlv.ac.uk (P.R.); eunzhang@liverpool.ac.uk (E.Z.); fabio.miyajima@liverpool.ac.uk (F.M.); munirp@liverpool.ac.uk (M.P.); 2Computational Biology Facility, Institute of Systems, Molecular and Integrative Biology, University of Liverpool, Liverpool L69 7BE, UK; aevans@liverpool.ac.uk; 3Clinical Directorate, GCP Laboratories, University of Liverpool, Liverpool L7 8TX, UK; 4Faculty of Science and Engineering, School of Biomedical Science and Physiology, University of Wolverhampton, Wolverhampton WV1 1LZ, UK; 5Oswaldo Cruz Foundation (Fiocruz), Branch Ceara, Eusebio 61773-270, Brazil

**Keywords:** *Clostridioides difficile*, transcriptomics, blood, adaptive immunity, immunometabolism

## Abstract

*Clostridioides difficile* (*C. difficile*) is a global threat and has significant implications for individuals and health care systems. Little is known about host molecular mechanisms and transcriptional changes in peripheral immune cells. This is the first gene expression study in whole blood from patients with *C. difficile* infection. We took blood and stool samples from patients with toxigenic *C. difficile* infection (CDI), non-toxigenic *C. difficile* infection (GDH), inflammatory bowel disease (IBD), diarrhea from other causes (DC), and healthy controls (HC). We performed transcriptome-wide RNA profiling on peripheral blood to identify diarrhea common and CDI unique gene sets. Diarrhea groups upregulated innate immune responses with neutrophils at the epicenter. The common signature associated with diarrhea was non-specific and shared by various other inflammatory conditions. CDI had a unique 45 gene set reflecting the downregulation of humoral and T cell memory functions. Dysregulation of immunometabolic genes was also abundant and linked to immune cell fate during differentiation. Whole transcriptome analysis of white cells in blood from patients with toxigenic *C. difficile* infection showed that there is an impairment of adaptive immunity and immunometabolism.

## 1. Introduction

*Clostridioides difficile* (formerly known as *Clostridium difficile* and commonly referred to as *C. difficile*) is an established threat to public health globally and a significant burden to healthcare systems [1,2,3]. Lately, there have been worries about its re-emergence [4] (Appendix A). The United Kingdom (UK), which has one of the lowest incidence rates in the world, has now had a 5-year consecutive increase (January 2019–April 2023) corresponding to a 10-year high [5]. This trend is echoed in reports from Australia and other countries [6,7]. Even in areas with decreasing health care-associated infections (US and Canada), recurrent and community-acquired infections appear to be on the rise [8,9,10,11,12]. Lack of centralized surveillance systems in Asia and Africa, reduced testing during the coronavirus-19 pandemic (COVID-19), and underdiagnosis mean that there is an underestimate of the true epidemiological burden globally [11,13,14,15].

*C. difficile* is a resilient, versatile pathogen. Over the years, it has evolved to develop various survival mechanisms, as reflected in its remarkable phylogenetic diversity, with sequenced isolates sharing less than one-fifth of genes [16]. Transmission sources are abundant given that *C. difficile* persists in both the environment and hosts; the spores are extremely difficult to eradicate and are responsible for (re-) initiating infection [17,18,19]. It is intrinsically resistant to most antibiotics without compromising fitness, and there is an emergence of strains with decreased susceptibility to vancomycin and metronidazole, which are two of three available anti-microbials for *C. difficile* [20,21,22]. In a US study, the majority of patients were healthy adults less than 65 years old [23], while a Swedish nationwide population-based study showed that mortality increased with infection irrespective of age and co-morbidity [24]. Asymptomatic carriage has been reported in around 3% of the adult population, but it can be as high as 10% in health-care settings [25]. Intestine pathogens facilitate *C. difficile* carriage and shedding [26,27], but colonization and symptomatic infection are indistinguishable with routine diagnostics, making management difficult [28].

Increased recurrence rates of *C. difficile* infection (~30% for a first episode) remain a major challenge—this increases mortality and subsequent relapses [29,30]. Fidaxomicin, which has narrow spectrum activity, has been shown to reduce recurrence rate, contrary to metronidazole, which has a wide anti-microbial burden that can have a deleterious effect on the gut microbiome [31,32]. The effectiveness of fecal microbiota transplant in recurrent but not primary infections underscores the importance of gut microbiome as a protective factor [33]. More recently, a microbiota-based oral therapeutic of Firmicutes spores (VOWST; VOS, formerly SER-109) has demonstrated promising results in clinical trials [34,35,36]. Another strategy to prevent recurrent infection is the use of monoclonal antibodies against *C. difficile* toxin B (bezlotoxumab) [37,38]. Patients who effectively produce neutralizing antibodies are at reduced risk of recurrence [39,40]. Hence, the development of a vaccine may be feasible. Currently, there is a candidate (PF-06425090) in late-phase clinical trials that does not prevent infection but reduces severity following a three-dosing (at least) regimen to achieve sustained immunity [41,42].

Overall, the interplay between gut dysbiosis and host immunity is complex and poorly investigated [43,44]. Thus, mechanistic understanding of *C. difficile* infection is important as it will help in developing better treatment and prevention strategies. Our study attempted to shed light on immune responses triggered by *C. difficile* at the gene expression level in peripheral blood. Assuming that gut dysbiosis and *C. difficile* infection, particularly, have an effect on the transcriptome of circulating cells, it is possible that this may result in a set of commonly expressed genes in patients with diarrhea due to different causes and a unique gene expression signature associated with *C. difficile* infection.

## 2. Results

### 2.1. Patient Characteristics

We collected clinical data and samples from 200 patients with diarrhea and 51 healthy controls (HC) across five sites in the Northwest of England between February 2013 and July 2015 (Appendix A). Patients infected with *C. difficile* were either cases, CDI (n = 78), or controls, GDH (n = 37), depending on the presence or absence of toxin, respectively. Of the 40 patients with inflammatory bowel disease (IBD), 67% (n = 27) had ulcerative colitis, 25% (n = 10) had Crohn’s disease and 8% (n = 3) had non-specific colitis. Diarrhea controls (DC, n = 45) comprised patients with infective (n = 16, 36%), antibiotic-associated (n = 8, 18%), and non-specific (n = 21, 47%) diarrhea (Figure 1).

Of the CDI patients, 62% (n = 48) had severe disease as per NICE criteria [45,46,47]. CDI patients were older with more co-morbidities, increased frailty, and more likely to have more than one acute illness compared with the IBD and HC groups (Table 1). IBD was the youngest group with the fewest co-morbidities but with the most severe symptoms in relation to the number of bowel movements and duration (Table 1). The diarrhea groups were inpatients, while HC were mainly (n = 28, 55%) attenders at hypertension, clinical biochemistry, and general internal medicine clinics (Appendix A). As a result, HC had a higher incidence of risk factors for cardiovascular disease (CVD), including high blood pressure, hyperlipidemia, obesity, smoking, and alcohol excess (Table 1).

The presenting complaints of all IBD patients were diarrhea and/or symptoms consistent with colitis (e.g., abdominal pain, per rectal bleeding), which led to early stool sampling and recruitment (Table 1). Most CDI and GDH patients had an acute illness which was followed by *C. difficile* infection (Table 1). The presence of entero-colic disease on admission was associated with earlier clinical stool samples and transcriptomic samples (Kruskal–Wallis test *p* < 0.001). There was no difference for the period between the transcriptomic sample and the worst day of diarrhea or antibiotic initiation among the groups (Table 1).

The distribution of white cell count (WCC) was the same across the groups, but if WCC was combined with temperature and creatinine to reflect severity, CDI patients had significantly more severe disease (Table 1). The overall mortality in the year following recruitment to this study was higher in patients with toxigenic and non-toxigenic *C. difficile* infection, with the CDI group having higher 28-day all-cause mortality (Table 1).

Half of the CDI (50%, n = 39) and more than half of the GDH (60%, n = 22) patients had a hospital-acquired infection, of which 67% (n = 26) and 73% (n = 16), respectively, had prior use of antibiotics. Around two-thirds of CDI and GDH patients were on a proton pump inhibitor (PPI) at the time of infection, and this figure was similar in DC (Table 1). Of the CDI and GDH stool samples, 68 (87%) and 20 (54%), respectively, were ribotyped: 002, 014, and 078 were the most frequent isolates (Appendix A). Toxin genes were detected with multiplex PCR in 61% of isolates (n = 11) from a subset (n = 18) of the GDH stool samples (Appendix A). Although clinical testing for *C. difficile* was negative (GDH-/CDT-) for the IBD and DC cohorts, stool cultures were positive in 5% (n = 2) and 16% (n = 7), respectively (Appendix A). Asymptomatic carriage was 2% (n = 1) amongst the HC.

### 2.2. Patients with Different Types of Diarrhea Have Significant Variation at the Gene Expression Level in Peripheral Blood Enriched for Innate Immunity and Neutrophil Activation

Principal component analysis (PCA) of the blood transcriptome showed that 29.7% of the total variance was represented by the first two principal components (Figure 2). Differential expression analysis identified a few thousand genes with a false discovery rate (FDR) adj. *p* < 0.05, indicating expression in diarrhea groups varied from HC (Figure 3 and Figure 4). One-fifth (n = 2255) of the differentially expressed genes were common in all diarrhea groups (Figure 5a), and the most significant reactome pathways were neutrophil degranulation and innate immune response (Figure 5b).

We narrowed down differentially expressed genes by applying an additional criterion of |log_2_FC| > 0.5. All diarrhea groups co-shared 362 genes, and neutrophil degranulation remained the most significantly upregulated pathway (Figure 4 and Appendix A). Downregulation of adaptive immunity appeared unique to CDI (Appendix A).

To search the literature, we confined differentially expressed genes between each diarrhea group and HC to the top 20 with the highest fold-change. Common genes in all comparisons were 12 (*ANKRD22*, *ANXA3*, *CD177*, *CEACAM8*, *DEFA4*, *GALNT14*, *HP*, *LTF*, *MGAM2*, *MMP8*, *OLFM4*, and *SLC26A8*) and they were all involved in neutrophil functions. A mini-review showed that transcripts from the common 12-gene set were found in the blood of patients with various inflammatory conditions, including sepsis and infections, cardiovascular disease, auto-immune diseases, cancer, mental illness, and pregnancy (Appendix A). Although we excluded studies with samples other than whole blood (e.g., PBMC and leucocytes) and non-adjusted *p*-values, we still encountered some of the common genes in non-eligible work (data not presented). Then, we focused on sepsis studies where members of the common 12-gene set were found in two studies (differentially expressed genes between sepsis endotypes) and three classifiers, two of which were derived from multi-cohort analysis (Appendix A). Moreover, the GEO database was searched to retrieve studies investigating the blood transcriptome in colitis (Appendix A), and we encountered genes from the 12-gene set in six of the seven eligible publications (Appendix A). The most popular genes in all our searches were *OLFM4*, *CEACAM8*, *HP,* and *MMP8* and the least frequent (found only in an IBD study) was *MGAM2*.

To investigate genes that were associated with severity, we modified the CDI criteria for severity and applied them to all the diarrhea groups (Appendix A). We found 487 differentially expressed genes with an FDR adj. *p* < 0.05 between severe vs. non-severe all-cause diarrhea cases, which were enriched for neutrophil degranulation and innate immunity. Of this 487-gene list, 236 were dysregulated significantly in severe CDI, and *SLPI* was amongst the top 20 upregulated. Of the common 12-gene set, 8 (67%) genes were significantly upregulated in patients with severe diarrhea compared with those with non-severe diarrhea (Appendix A).

A subset of CDI patients was sampled 2 weeks following infection (n = 25). Due to the small sample size, we used gene-sets for differential expression analysis. Most genes (n = 10) of the common 12-gene set showed a reduction in expression from baseline (Appendix A), and only *SLPI* in the severity 487-gene set was significantly (FDR adj. *p* < 0.05) decreased in repeat sampling.

IBD was the only group where transcriptomics in blood had been performed previously. Of the seven studies we found through the GEO database (Appendix A), three presented differentially expressed genes between patients with ulcerative colitis and/or Crohn’s disease and healthy controls. We extracted genes with an FDR adj. *p* < 0.05 and |log_2_FC| > 0.5 (or |FC| > 1.5) from these studies. There was an overlap with our cohort (1769 differentially expressed genes between IBD and HC with an FDR adj. *p* < 0.05 and |log_2_FC| > 0.5), as shown in Figure 6.

### 2.3. CDI Is Associated with Dysregulation of Adaptive Immunity

We compared CDI with the three diarrhea groups (GDH, IBD, and DC), and we found that gene expression in peripheral blood differed, with most genes being downregulated in CDI (Figure 7a–c). Although neutrophil degranulation was observed between CDI vs. DC, it did not appear when CDI was compared with IBD or GDH (Figure 7d–f). A comparison of CDI vs. all four controls with Ingenuity Pathway Analysis (IPA) revealed that the IL-8 pathway was the only significant canonical pathway (|z-score| > 2) in all comparisons, which was least upregulated in CDI compared with the control groups (Figure 8a).

We identified 45 genes that were uniquely dysregulated in CDI compared with all four control groups, and all but three were downregulated (Figure 8b). Gene set enrichment analysis (GSEA) did not return any enriched terms with any resource (Reactome and Hallmark database via R studio, Spring, DAVID). We checked the 45 gene list in severe (n = 48) vs. non-severe (n = 30) CDI cases (as defined by NICE) and in the subset of baseline vs. repeat samples (n = 25), and there were no genes with an FDR adj. *p* < 0.05.

Functions were, therefore, manually curated and summarized for each gene of the 45 unique signature (Table 2). Over one-third of genes (n = 16) were involved in mitochondrial function with downregulation of factors participating in fatty acid oxidation and mitophagy (mitochondrial autophagy). Almost a third of genes (n = 13) were associated with T-cell memory and humoral immunity. The rest had important roles in metabolism and the cytoskeleton or were poorly described in the literature. There were two transcripts, *GPAA1* and *PIGU*, which were directly associated as subunits for the glycosylphosphatidylinositol transamidase (GPI-T) complex. GPI-T is composed of five subunits (*GPAA1*, *PIGU*, *PIGS*, *PIGT,* and *PIGK*) [51], with *PIGS* and *PIGT* also being significantly downregulated in CDI and IBD (vs. HC).

## 3. Discussion

This is the first report of altered gene expression in peripheral blood during *C. difficile* infection. We identified an innate immune response, where neutrophils played a central role, shared by all diarrhea groups, whereas colitis due to toxigenic *C. difficile* infection was characterized by the downregulation of genes associated with adaptive immunity.

As expected, our groups of patients were remarkably heterogeneous. The *C. difficile* infection (toxigenic and non-toxigenic) cohort was typical, mostly older than 65 years and frail with many co-morbidities, had prolonged hospitalization with an acute illness, and demonstrated increased mortality [9,45]. Subjects with IBD were in their 40s, mildly frail, and had prolonged diarrhea before presentation and sampling. The DC group laid in the middle with advanced age and many co-morbidities but was generally less frail and had shorter hospitalization before recruitment compared with the CDI and GDH groups.

Despite the heterogeneity between groups, we identified a common inflammatory response at the transcription level in blood. Innate immunity and particularly neutrophil-related functions were significantly upregulated in all diarrhea groups compared with our HC, which was consistent with previous reports [48]. This non-specific response in the blood was shared in most inflammatory conditions, including sepsis. Presumably, there were more than 12 (that our search was confined to) co-shared genes from the big pool of common genes (2255). Circulating neutrophil subpopulations, although they have lower total RNA content per cell compared with other circulating cells, alter their transcriptome to minor environmental changes such as brief exercise, and this enhancement is proportional to the extent of inflammation/disease and migration activity [137,138,139,140]. Nevertheless, the literature is scarce regarding the overlap in neutrophil activation signatures amongst the different inflammatory conditions [141,142].

Among the commonly encountered genes during inflammation, *MGAM2,* which encodes maltase-glucoamylase 2, was reported in a study in inflammatory bowel disease as a member of a multi-gene set in blood reflecting inflammation in the gut [143]. It was upregulated in severe cases but did not significantly decrease in the two weeks after the onset of *C. difficile* infection. *MGAM2* is largely uncharacterized and potentially associated with immune responses and glucose metabolism [144,145]. *SLPI*, on the other hand, was the only gene that was significantly upregulated in severe cases and downregulated in repeat sampling. *SLPI* is an anti-microbial peptide with anti-inflammatory properties secreted by epithelial and immune cells [146,147]. Its high expression in various tumors has been inconsistently linked to prognosis depending on the cancer stage [148,149,150]. Moreover, it is a principal regulator of gut responsiveness to microbial stimuli by ameliorating the destructive effect of inflammation [151]. Colonic recruitment of monocytes and neutrophils during experimental *C. difficile* infection in mice has been proposed to be a major producer of SLPI [152,153].

Our CDI cohort uniquely expressed a 45 gene signature. Although the 45 unique genes were not computationally enriched, we identified important patterns through analysis of the literature. Most genes in CDI play important roles in adaptive immunity and cell metabolism, particularly fatty acid oxidation and mitochondrial homeostasis. Downregulation of immune-related genes such as *CD19* (B cell-specific marker), *SPIB* (B cell memory), *STING1* (T cell memory), *GPA33* (memory CD4), *VHL* (memory CD8), and *LPAR5* (effector lymphocytes) potentially points towards an impairment in antibody production and differentiation into memory cells.

In concert, metabolic reprogramming and its principal regulators, the mitochondria, directly influence immune cell proliferation, differentiation, homeostasis, and recall responses [154,155]. Hence, reduced expression of genes involved in the critical steps of catabolism such as fatty acid oxidation and subsequent oxidative phosphorylation (*COQ10A*, *CYB561A3*, *NDUFAF3*, *PEX3*, *SMIM20*, *ALDH9A1*, and *COQ6*), and upregulation of facilitators of rapid energy production (*NSUN7*, *VHL*) portray an active effector phenotype of immune cells rather than quiescent memory. Moreover, disrupted autophagy in memory lymphocytes results in the accumulation of damaged mitochondria, disrupted fatty acid oxidation, and attenuated secondary immune responses [154,156]. This was observed in our CDI unique signature by downregulation of genes maintaining mitochondrial integrity (*CYB561A3*, *NAA30*, *PMPCA*, and *TOP1MT*) and promoting mitophagy (*EI24*, *FANCF*, and *RNF41*) or autophagy (*GPAA1*, *PIGU*, and *SPIB*).

Immunometabolism is a relatively recent concept but also a rapidly evolving field interlinked with oxidation–reduction (redox) reactions [157,158]. In the colonic mucosa of mice infected with *C. difficile*, upregulation of glycolytic metabolism has been observed during the peak of inflammation which reverted to mitochondrial metabolism during the recovery phase [159]. Key players in lymphocyte fate and metabolism are mitochondria [160] and mitochondrial autophagy (or mitophagy) is necessary for maintaining long-lived IgG memory B cells [161]. This could be crucial in the generation of humoral immunity. Indeed, the production of neutralizing antibodies against *C. difficile* toxins has been shown to be ineffective in various reports [162,163]. This is also reflected in the increased recurrence rate and low immunogenicity [164,165]. T cell-mediated immunity is even more elusive, but evidence suggests that CD4 cells may have an important role in *C. difficile* clearance and reversal of gut microbiome homeostasis [43,166].

Our unique CDI signature included genes associated with the cytoskeleton and genomic stability, which appear to have important roles in cell homeostasis. Cytoskeleton components play important roles in lymphocyte polarization and communication through the immunological synapse, as well as receptor signaling [72,167,168]. Similarly, DNA damage repair systems are crucial to adaptive immunity for receptor diversity (TCR/BCR) and antibody specificity (V(D)J recombination, class switch recombination, and somatic hypermutation) [169].

Interestingly, some of the 45 unique genes could be mediators of *C. difficile* pathogenesis or targets of toxin A (TcdA) and B (TcdB). TcdA binds to sulfated glycosaminoglycans, which are found on GPI-anchored receptors (*GPAA1*, *PIGU*), while low-density lipoprotein receptor (*LDLRAP1*) assists cell entry [170]. TcdB binds frizzled receptors inhibiting the Wnt/β-catenin pathway (*OLFM4*, *STING1*), which is required for the generation of CD8 memory stem cells [98,171,172]. Both toxins inactivate small Rho GTPases by glycosylation, resulting in cytoskeletal disruption, cell rounding, cell cycle arrest, and death [173]. Moreover, they induce preliminary Ca^2+^ influx, disrupt mitochondrial integrity, and generate reactive oxygen species (ROS) with subsequent redox (*MOCS1*) signaling and DNA damage [174,175]. *C. difficile* toxins cause mitochondrial damage in the colonic epithelium [176]. It is possible that circulating immune cells are infected with *C. difficile* toxins, but their effect at the molecular level can only be speculated [177,178].

Our study has several limitations. First, this is a discovery cohort, and we have not performed clinical or technical validation. We regrettably report that top genes have not been confirmed with quantitative or droplet PCR due to inadequate sample volume. As this is the first report of transcriptomic changes in peripheral blood in patients with *C. difficile* infection, the lack of publicly available datasets prevented in silico validation. Therefore, data should be used cautiously in the future in consideration of the lack of validation; however, it is reassuring from the experimental perspective that common and IBD-related genes were replicated in other work. Second, the findings are based on gene expression products, and we have not explored proteins or proteomics. Functions have been inferred from the protein end-product and even though mRNA and protein levels are correlated in the literature, we have not performed confirmatory experiments in our cohort. Nevertheless, links among CDI-unique genes provide compelling avenues for further investigation. Third, we explored bulk RNA in blood which is justified given that this topic has not been previously approached in the literature. The findings question our understanding of peripheral host immune responses to toxigenic *C. difficile* and offer directions for further study with more precise and accurate methods, such as single-cell RNA-sequencing in blood and spatial transcriptomics in colonic tissue.

## 4. Materials and Methods

### 4.1. Study Population

Patients with diarrhea were recruited from five hospitals (Royal Liverpool Hospital or RLH, University Hospital of Aintree or UHA, St. Helen’s and Knowsley or STK, Whiston Trust Hospital or WTH and Walton Centre for Neurology and Neurosurgery or WNN) in the Northwest of England. Patients who developed diarrhea that led to or during hospital admission and were categorized between grades five and seven on the Bristol stool chart were allocated into four groups: CDI, GDH, IBD, and DC. Stool samples were tested for glutamate dehydrogenase (GDH) and *C. difficile* toxin (CDT) A and/or B and/or binary toxin with a two-step protocol [179] at the local clinical microbiology laboratory. CDI cases were toxin-positive with a double positive test (GDH+/CDT+), and diarrhea controls were toxin-negative (Figure 1). Patients with previously confirmed *C. difficile* infection were excluded. We also recruited healthy volunteers (HC).

Severe CDI cases were defined as per the NICE and European criteria for *C. difficile* infection [46,47]. We used a modification of these criteria to apply to all diarrhea groups. Severe cases were defined if they had one of (1) white cell count (WCC) > 15 × 10^9^ cells/L, (2) serum creatinine > 133 μmol/L, and (3) temperature > 38.5 °C (Appendix A).

### 4.2. Sample Collection and Processing

Stool and blood samples were collected at the time of recruitment (baseline) for all subjects and two weeks following the diagnosis of *C. difficile* infection (repeat) for a subset of CDI patients.

Stool samples with sufficient yield following routine testing were anonymized and stored at −80 °C. All fecal samples underwent alcohol-shock treatment, followed by a culture for *C. difficile* on Brazier’s cefoxitin–cycloserine egg yolk agar (EO Labs, Bonnybridge, United Kingdom). Internal core sequences of both toxins A and B genes and a species-specific internal fragment of the triose-phosphate isomerase (TPI) housekeeping gene were targeted by a multiplex PCR assay in control samples, as previously described [180].

Whole blood samples were collected in Tempus Blood RNA Tubes (Thermo Fisher Scientific, Loughborough, UK) and stored at −80 °C. RNA was purified with the Tempus Spin RNA Isolation kit (Applied Biosystems, Foster City, USA), which utilizes a glass fiber filter-based technique for RNA isolation as per the manufacturer’s instructions. The quality control (QC) assessment of extracted RNA was performed with the Agilent 2100 Bioanalyzer and Thermo Scientific NanoDrop Spectrophotometer. Samples with RNA integrity number (RIN) ≥ 6.5 (Appendix A) and RNA minimum quantity of 50 ng were processed with the Clariom^TM^ D assay for Humans on the GeneChip console (Affymetrix, Singapore ) at the European (Nottingham) Arabidopsis Stock Centre (NASC) as per manufacturer protocol (GeneChip WT PLUS Reagent Kit User Guide (P/N 703174, Rev A.0, Thermo Fisher Scientific Inc.).

### 4.3. Microarray Data Processing

All microarray data preprocessing and analyses were performed in R version 4.4.2. The code to perform the R analyses in this paper is available at “https://github.com/CBFLivUni/Gene_expression_dysregulation_in_whole_blood_of_patients_with_C_difficile_infection.git (accessed on 21 November 2024)”.

Clariom D files were imported, and log_2_-transformed raw data were inspected for outliers using relative log expression plots. One DC sample was excluded from further analysis as all probe intensity values were considerably lower than the remaining microarrays, with many below the limit of detection. Background subtraction, quantile normalization, and summarization were performed using the robust multichip average algorithm at the ‘core’ target level, as implemented by the ‘oligo’ R package.

Prior to statistical analysis, a background intensity threshold was chosen based on histograms of median transcript cluster (TC) intensity across all samples and background control probe sets. TCs with at least 37 samples—the smallest disease group set—possessing greater intensity than the threshold were included in the analyses.

TCs were annotated with gene IDs using the ‘AnnotationDbi’ package. Those that mapped to multiple Entrez gene IDs were excluded, and where multiple TCs mapped to the same gene ID the TC with the greatest mean expression across samples was selected and the remainder filtered out.

### 4.4. Differential Expression Analyses

Tests for differences in mean RNA abundance between disease groups were performed using the ‘limma’ R package, adjusting for age and sex. To allow for potential non-linear associations between age and gene expression, age was fitted as a natural spline with 3 degrees of freedom. The potential effects of the microarray batch were assessed by principal component analysis (Appendix A). No clear association was apparent, and batch ID was not adjusted for in the differential expression analyses.

*The p*-values were adjusted for FDR using the Benjamini–Hochberg method, with gene contrasts yielding adjusted *p*-values < 0.05 considered to be differentially expressed.

### 4.5. Gene Set Enrichment Analysis

GSEA was performed ranking genes by the mean of t-statistics from differential expression analyses of each diarrhea group vs. HC and CDI vs. each control group using the fgsea package [181], with Hallmark and Reactome gene set downloaded from “https://www.gsea-msigdb.org/gsea/msigdb/collections.jsp (accessed on 3 August 2022)”.

### 4.6. Ingenuity Pathway Analysis

Common genes: Diarrhea common genes were identified through Venn diagrams at the area of convergence from the differential expression analysis (FDR adj. *p* < 0.05) of each diarrhea group vs. the HC arm. To narrow down the number of common genes, filtering criteria were applied, including an absolute value of the logarithm with base 2 of fold change (|log_2_FC|) > 0.5 and the top 20 dysregulated genes with the highest |log_2_FC|.Unique genes: CDI unique genes were identified through Venn diagrams at the area of convergence of differential expression analysis (FDR adj. *p* < 0.05) of CDI cases vs. each control group (GDH, IBD, DC, and HC).Severity genes: Diarrhea severity genes derived from differential expression analysis (FDR adj. *p* < 0.05) between severe vs. non-severe diarrhea (combined CDI, GDH, IBD, and DC) as defined with the modified criteria.

The IPA by Qiagen, the String v12.0 database (https://string-db.org/), and the Database for Annotation, Visualization, and Integrated Discovery (DAVID) were used for functional annotation of gene sets.

### 4.7. Literature Search

To answer the question if the common diarrhea gene set is specific to entero-colitis or encountered in other illnesses, the literature was searched for the top common genes from three different aspects: mini-review of the literature, blood transcriptomics in sepsis, and blood transcriptomics in colitis.

For the non-exhaustive mini-review, we conducted PubMed searches with two gene-symbol combinations from the diarrhea common signature. Studies were included if they had at least five subjects per group, whole genome expression in whole blood was investigated with commercial microarrays or RNA-sequencing, and differential gene expression analysis between cases and healthy controls used an adjusted *p*-value < 0.05. Only data from discovery cohorts were investigated. If a list of differentially expressed genes was not published, we selected important genes from the data presented in the publication. Important genes were hub or crucial genes suggested by the authors and/or genes that ranked highly with annotation tools such as protein-protein interactions (PPI) networks and IPA [50,182,183,184,185,186,187,188,189,190,191,192,193,194,195,196,197,198,199,200,201,202,203,204,205,206,207,208,209,210,211,212,213,214,215,216,217,218,219,220,221].

Sepsis is probably a field in which gene expression signatures will probably be translated into clinical practice in the near future. We searched sepsis studies, as previously summarized by Tsakiroglou et al. [222], for the top common diarrhea genes [223,224,225,226,227]. Finally, we also conducted a Gene Expression Omnibus (GEO) database search for colitis studies that have reported from the top common diarrhea gene set (Appendix A) [48,49,50,143,228,229,230].

To functionally annotate the CDI unique gene set, we searched PubMed and the European Bioinformatics Institute (EBI) Expression Atlas with the gene symbols and manually curated the most relevant information.

## 5. Conclusions

*C. difficile* is an interesting pathogen because of its resilience (extreme spore resistance), adaptation (remarkable phylogenetic variability, pronounced antimicrobial resistance, toxigenic and non-toxigenic isolates), and relationships (unique host and gut microbiome equilibrium, non-pathogenic colonization at an early age and asymptomatic carriers, increased risk for relapse). These unique characteristics, in concert with the lack of adequate therapeutics, impose a global threat and underscore the importance of understanding molecular mechanisms of host immune responses. We propose that future research with a focus on adaptive immunity and immunometabolism will fuel the discovery of novel therapeutic targets for *C. difficile*. Genes such as *MGAM2* and *SLPI* require further characterization as biomarkers for patient stratification. Last but not least, the co-expression of neutrophil-related genes during various inflammatory conditions is a concept that warrants further investigation.

## Figures and Tables

**Figure 1 ijms-25-12653-f001:**
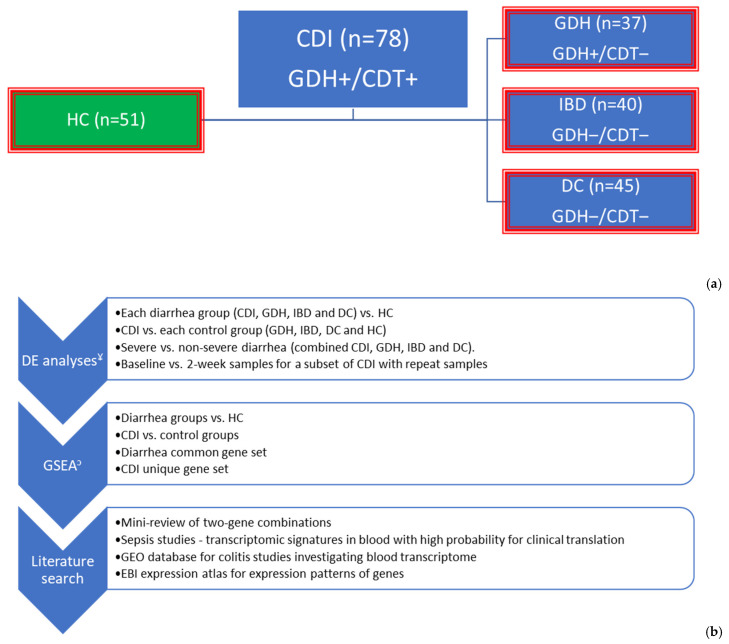
Graphical summary of methodology: (i) To identify diarrhea common immune responses in peripheral blood, each diarrhea group (blue boxes) was compared with HC (green box), and the overlapped genes in all four differential expression analyses were extracted. (ii) To identify CDI unique genes, CDI was compared with each of the control groups (red outline), and the overlapped genes in all four differential expression analyses were extracted. (iii) Functional annotation of differentially expressed genes and gene sets was performed with GSEA and literature searches. (**a**) Cases and controls and (**b**) Analysis plan. ^¥^ The primary filter for DE analysis was FDR adj. *p* < 0.05. When gene-sets were required to have a maximum number for GSEA (e.g., String database 3000 proteins max), |log_2_FC| < 0.5 was added. To reduce further the number of genes for literature searches, we used the top 20 genes with the highest |log_2_FC|, ^ↄ^: GSEA was performed using R studio (Reactome pathways), IPA (summary and comparison), and String online database. DE: differential expression, CDI: toxigenic *C. difficile* infection, GDH: non-toxigenic *C. difficile* infection, IBD: inflammatory bowel disease, DC: diarrhea controls, HC: healthy controls, NICE: National Institute for Health and Care Excellence, GSEA: gene set enrichment analysis, EBI: European Bioinformatics Institute, FDR adj. *p*: false discovery rate adjusted *p*-value, |log_2_FC|: an absolute value of logarithm with base 2 of fold change, IPA: Ingenuity Pathway Analysis by QIAGEN.

**Figure 2 ijms-25-12653-f002:**
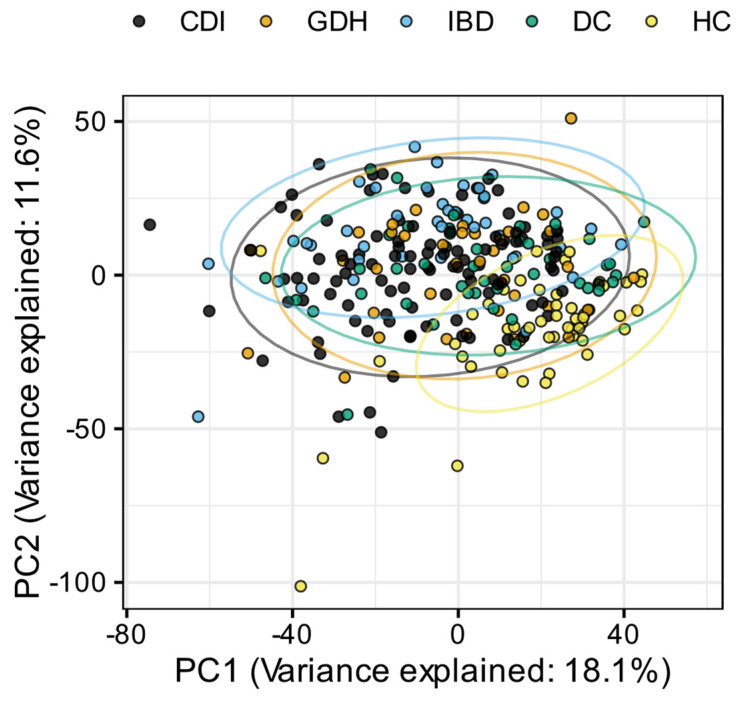
Principal component analysis (PCA) normalized gene expression data for each sample, colored by participant groups. CDI: toxigenic *C. difficile* infection, GDH: non-toxigenic *C. difficile* infection, IBD: inflammatory bowel disease, DC: diarrhea controls, HC: healthy controls, PC1: primary component 1, PC2: primary component 2.

**Figure 3 ijms-25-12653-f003:**
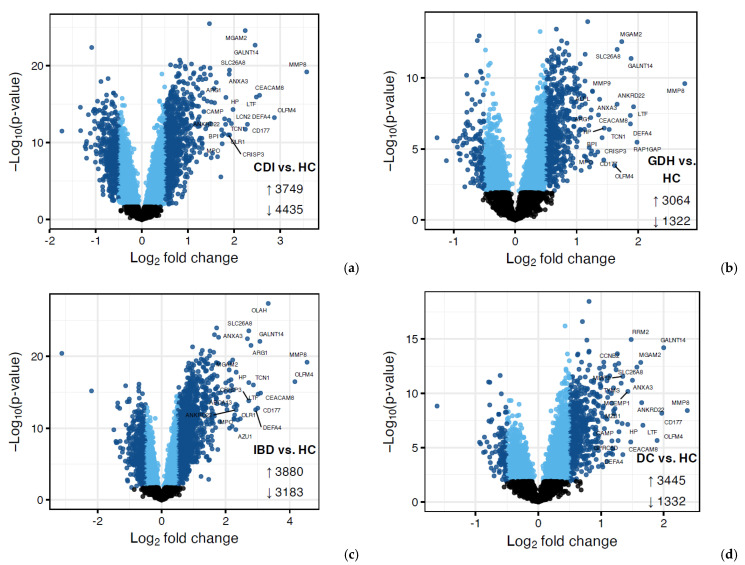
Volcano plots (**a**–**d**) of differential expression (DE) analysis of diarrhea groups vs. HC where genes with FDR adjusted *p* < 0.05 are displayed in blue, with darker blue points indicating those with |log_2_FC| > 0.5. CDI: toxigenic *C. difficile* infection, GDH: non-toxigenic *C. difficile* infection, IBD: inflammatory bowel disease, DC: diarrhea controls, HC: healthy controls, FDR adj. *p*: false discovery rate adjusted *p*-value, |log_2_FC|: logarithm with base 2 of fold change.

**Figure 4 ijms-25-12653-f004:**
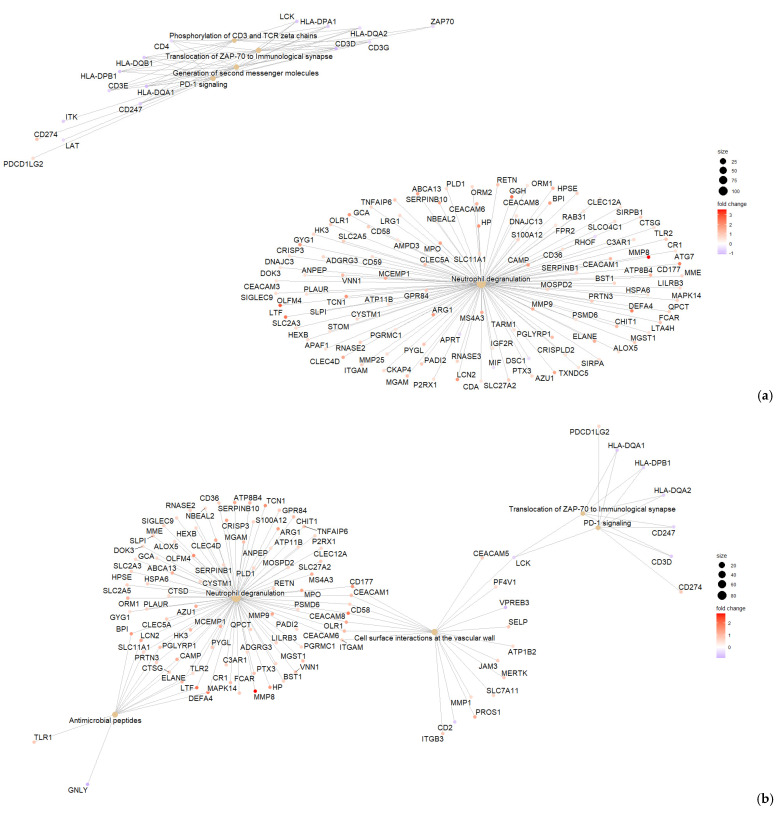
Enrichment summaries of differentially expressed genes with FDR adjusted *p* < 0.05 and|log_2_FC| > 0.5 between CDI vs. HC (**a**), GDH vs. HC (**b**), IBD vs. HC (**c**) and DC vs. HC (**d**) developed with IPA. CDI: toxigenic *C. difficile* infection, GDH: non-toxigenic *C. difficile* infection, IBD: inflammatory bowel disease, DC: diarrhea controls, HC: healthy controls, FDR adj. *p*: false discovery rate adjusted *p*-value, |log_2_FC|: logarithm with base 2 of fold change, IPA: ingenuity pathway analysis.

**Figure 5 ijms-25-12653-f005:**
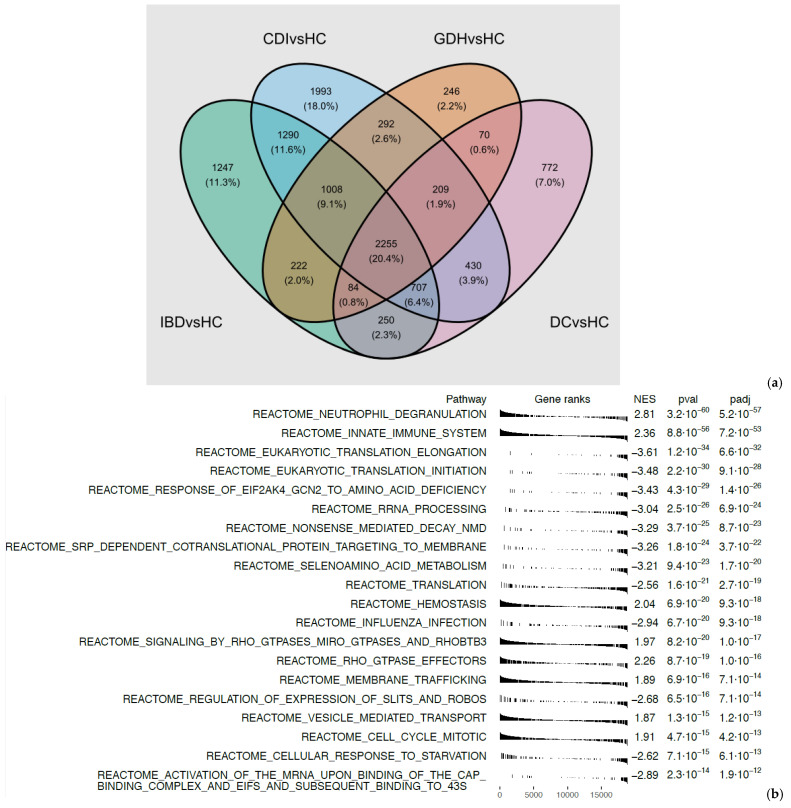
(**a**) Venn diagram of diarrhea groups vs. HC (FDR adj. *p*-value < 0.05) and (**b**) Reactome pathways from GSEA of pooled t statistics comparing all diarrhea groups vs. HC. CDI: toxigenic *C. difficile* infection, GDH: non-toxigenic *C. difficile* infection, IBD: inflammatory bowel disease, DC: diarrhea controls, HC: healthy controls, FDR adj. *p*: false discovery rate adjusted *p*-value.

**Figure 6 ijms-25-12653-f006:**
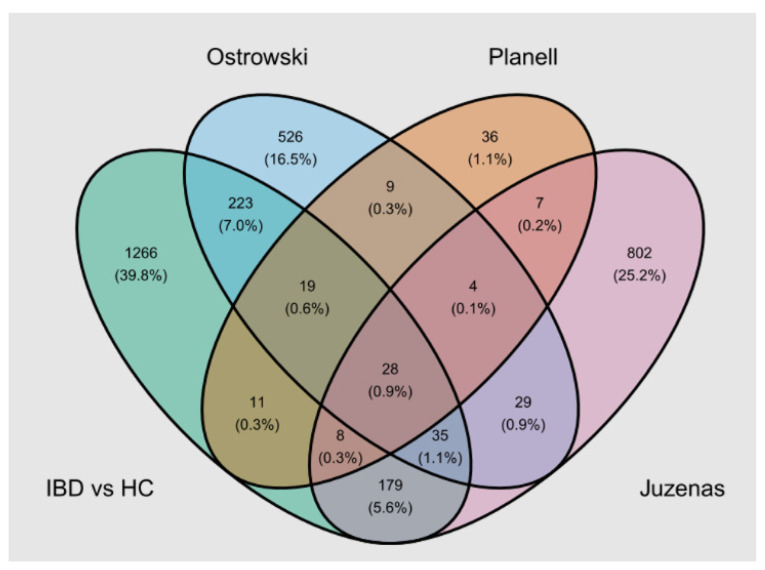
Venn diagram representing the overlap of differentially expressed genes between IBD and HC (FDR adj. *p*-value < 0.05 and |log_2_FC| > 0.5) among our cohort (IBD vs. HC) and published studies [48,49,50]. IBD: inflammatory bowel disease, HC: healthy controls.

**Figure 7 ijms-25-12653-f007:**
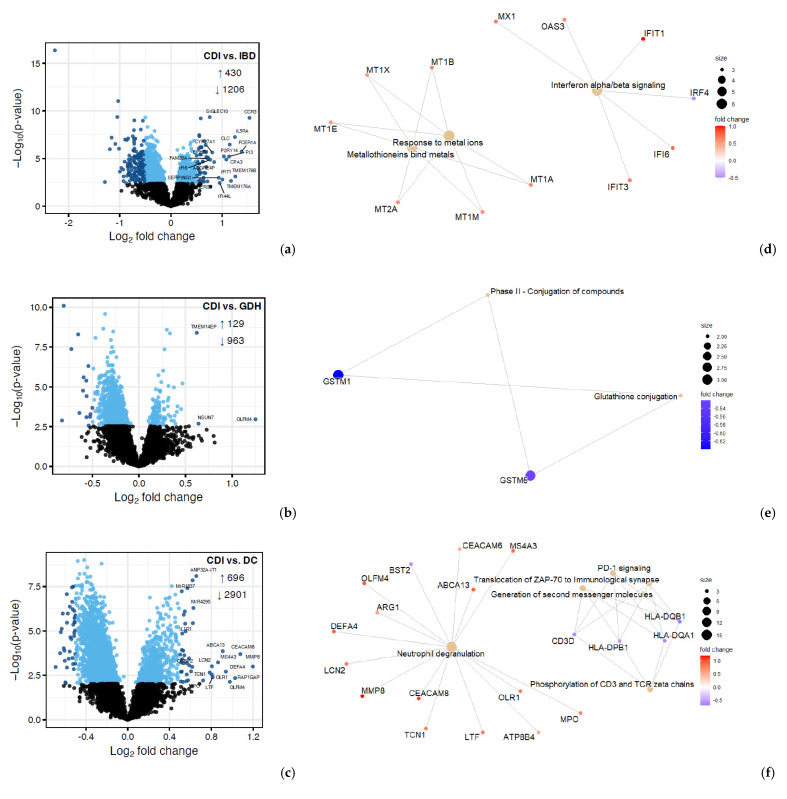
Differential expression (DE) analysis of CDI vs. diarrhea groups. (**a**–**c**) Volcano plots where genes with FDR adjusted *p* < 0.05 are displayed in blue, with darker blue points indicating those considered differentially expressed for IPA. (**d**–**f**) Enrichment summaries of genes with |log_2_FC| > 0.5. CDI: toxigenic *C. difficile* infection, GDH: non-toxigenic *C. difficile* infection, IBD: inflammatory bowel disease, DC: diarrhea controls, FDR adj. *p*: false discovery rate adjusted *p*-value, |log_2_FC|: logarithm with base 2 of fold change, IPA: Ingenuity Pathway Analysis.

**Figure 8 ijms-25-12653-f008:**
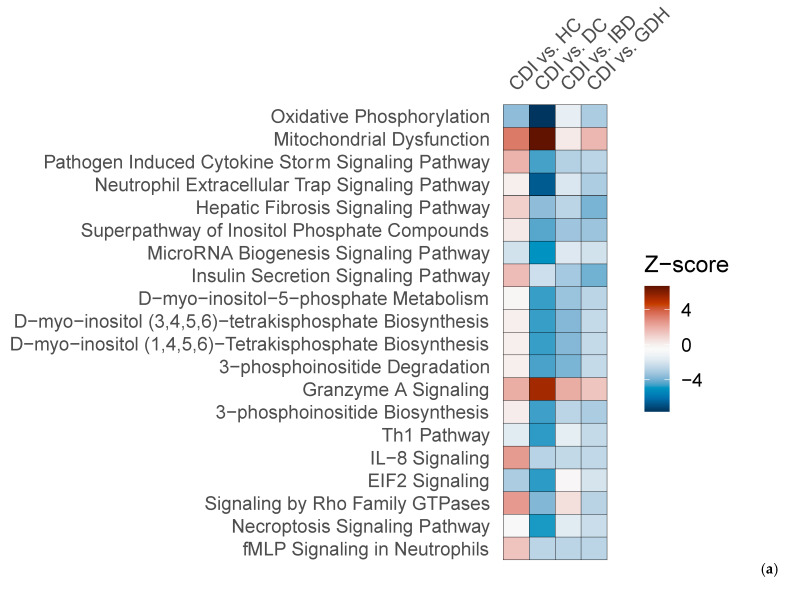
(**a**) Top 20 canonical pathways of the IPA comparison of CDI vs. HC, CDI vs. DC, CDI vs. IBD, and CDI vs. GDH (|z-score| > 2) and (**b**) Venn diagram of CDI vs. all control groups (FDR adj. *p*-value < 0.05). CDI: toxigenic *C. difficile* infection, GDH: non-toxigenic *C. difficile* infection, IBD: inflammatory bowel disease, DC: diarrhea controls, HC: healthy controls, FDR adj. *p*: false discovery rate adjusted *p*-value.

**Table 1 ijms-25-12653-t001:** Clinical variables per study group.

Clinical Variable	CDI (n = 78)	GDH (n = 37)	IBD (n = 40)	DC (n = 45)	HC (n = 51)	*p*-Value ^1^
Age, median years (IQR)	76 (65–83)	73 (57–82)	40 (28–66)	69 (61–76)	65 (62–72)	<0.001
Gender, male n (%)	35 (45)	14 (38)	21 (53)	21 (47)	29 (57)	0.434
Ethnicity, white n (%)	77 (99)	37 (100)	39 (98)	44 (98)	50 (98)	0.908
BMI, median (IQR)	24 (21–28)	25 (22–31)	25 (23–28)	27 (22–33)	28 (23–31)	0.054
Charlson comorbidity index, median (IQR)	5 (3–6)	4 (3–5)	0 (0–2)	4 (2–5)	3 (2–3)	<0.001
Comorbidities, n (%)						
Diabetes	22 (28)	12 (32)	2 (5)	14 (31)	6 (12)	0.003
Hypertension	39 (50)	23 (62)	7 (18)	23 (51)	22 (43)	0.001
Hyperlipidemia ^2^	43 (55)	19 (51)	7 (18)	26 (59)	27 (53)	<0.001
CVD ^3^	49 (63)	20 (54)	8 (20)	26 (58)	16 (31)	<0.001
Respiratory disease ^4^	21 (27)	8 (22)	13 (33)	17 (38)	11 (22)	0.357
CKD	25 (32)	8 (22)	1 (3)	7 (16)	1 (2)	<0.001
GI disease ^5^	18 (23)	11 (30)	40 (100)	16 (36)	0 (0)	<0.001
Malignancy	14 (18)	7 (19)	1 (3)	9 (20)	4 (8)	0.058
Auto-immune disease	9 (12)	4 (11)	40 (100)	7 (16)	1 (2)	<0.001
Immunosuppression	2 (3)	0 (0)	7 (18)	2 (4)	1 (2)	0.001
Obesity (BMI > 30)	19 (24)	11 (30)	6 (15)	15 (33)	17 (33)	0.192
Current smoker	9 (12)	3 (8)	7 (18)	10 (22)	5 (10)	0.198
Alcohol excess ^6^	5 (6)	5 (14)	34 (14)	5 (11)	11 (22)	0.002
Frailty ^7^, n (%)						<0.001
Mild	21 (27)	13 (35)	10 (25)	13 (29)	12 (24)	
Moderate	9 (12)	6 (16)	1 (3)	7 (16)	1 (2)	
Severe	36 (46)	16 (43)	4 (10)	4 (9)	1 (2)	
Carer aid, median hours/week (IQR)	0 (0–52)	0 (0–20)	0 (0–0)	0 (0–0)	0 (0–0)	<0.001
Health state ^8^, median (IQR)	50 (31–60)	50 (30–70)	60 (50–70)	63 (41–80)	80 (75–90)	<0.001
Primary diagnosis, n (%)						<0.001
Entero-colitis	29 (37)	9 (24)	40 (100)	24 (53)	-	
Other GI pathology	5 (6)	3 (8)	-	5 (11)	-	
Respiratory disease	4 (5)	1 (3)	-	4 (9)	-	
Cardiovascular disease	8 (10)	5 (14)	-	6 (13)	-	
Infection	10 (13)	8 (22)	-	3 (7)	-	
Other	9 (12)	5 (14)	-	2 (4)	-	
Elective admission	6 (8)	1 (3)	-	-	10 (20)	
Transfer	7 (9)	4 (11)	-	1 (2)	-	
No admission	-	1 (3)	-	-	41 (80)	
Presence of entero-colitis on admission, n (%)	29 (37)	9 (24)	40 (100)	24 (53)	-	<0.001
Number of stool motions on worst day, median (IQR)	5 (4–8)	4 (2–6)	10 (6–13)	5 (3–8)	NA	<0.001
Acute illness other than entero-colitis ^9^, n (%)	48 (62)	25 (68)	1 (2.5)	24 (53)	7 (18)	<0.001
Days from admission to clinical stool sample, median (IQR)	3 (1–12)	5 (2–12)	1 (1–3)	1 (1–3)	NA	<0.001
Days from admission to transcriptomic sample, median (IQR)	7 (4–16)	10 (7–16)	4 (3–6)	4 (3–6)	4 (3–4)	<0.001
Days from the onset of diarrhea to transcriptomic sample, median (IQR)	6 (4–10)	6 (4–11)	11 (8–19)	4 (3–6)	NA	<0.001
Days from worst day to transcriptomic sample, median (IQR)	3 (2–4)	3 (2–4)	2 (1–4)	2 (2–3)	NA	0.394
Days from the start of antibiotics to transcriptomic sample, median (IQR)	6 (4–6)	5 (2–17)	5 (3–7)	5 (3–6)	NA	0.852
Transfusion within 5 days from transcriptomic sample, n (%)	4 (5)	5 (14)	0 (0)	3 (7)	0 (0)	0.025
Highest NEWS, median (IQR)	2 (1–3)	2 (1–3)	1 (0–2)	2 (1–4)	NA	0.094
Maximum temperature median °C (IQR)	37.3 (37–38)	37.2 (36.9–37.6)	37 (36.8–37.3)	37.2 (36.9–37.8)	36.7 (36.6–36.8)	<0.001
Lowest BP (mmHg), median (IQR)	100/58 (89/52–110/65)	95/60 (86/50–115/64)	108/64 (100/60–120/70)	100/60 (92/55–110/66)	138/81 (124/73–158/89)	<0.001
WCC (×10^9^/L), median (IQR)	14.5 (10.1–18.8)	12.1 (9.9–15.1)	12.4 (9.5–14.8)	11.6 (9.2–15.9)	NA	0.100
CRP (mg/dL), median (IQR)	111 (52–194)	82 (36–144)	52 (13–92)	47 (23–195)	NA	0.006
PLT (×10^9^/L), median (IQR)	253 (199–352)	321 (247–401)	333 (244–436)	227 (171–314)	NA	<0.001
Creatinine (μmol/L), median (IQR)	104 (70–140)	80 (50–119)	75 (66–96)	86 (67–131)	NA	0.006
Albumin (mg/dL), median (IQR)	32 (28–36)	31 (25–37)	39 (34–41)	38 (35–42)	NA	<0.001
Severe diarrhea ^10^, n (%)	66 (85)	15 (41)	11 (28)	23 (51)	NA	0.004
Mortality, n (%)						<0.001
28-days	7 (9)	0 (0)		1 (2)		
90-days	5 (6)	2 (5)		0 (0)		
1-year	11 (14)	7 (19)		3 (7)		
Overall	23 (29)	9 (24)	0 (0)	4 (9)	0 (0)	
Colectomy within a year from recruitment, n (%)	1 (1)	0 (0)	8 (5)	0 (0)	0 (0)	<0.001
Prior PPI use, n (%)	49 (63)	28 (76)	17 (43)	34 (76)	12 (24)	<0.001

^1^ *p*-value derived from Kruskal–Wallis or Pearson chi-square test for continuous or categorical variables, respectively; ^2^: hyperlipidemia also includes patients with fatty liver disease and/or on lipid lowering treatment; ^3^: CVD as per WHO including coronary artery disease (myocardial infraction, angina, atrial fibrillation, heart failure, pacemaker, structural heart disease e.g., hypertrophy), cerebrovascular disease (stroke/transient ischaemic attack), peripheral arterial disease, deep venous thrombosis or pulmonary embolism, rheumatic and congenital heart disease; ^4^: respiratory disease includes chronic obstructive pulmonary disease, bronchiectasis, fibrosis, asthma, recurrent infections or tuberculosis within the last 5 years; ^5^: GI disease includes diverticulosis, coeliac, liver failure, Hirschsprung disease, chronic pancreatitis; ^6^: alcohol excess was defined as alcohol intake of 14 units and above; ^7^: Frailty was categorized based on mobility (A: no problems walking, B: some problems walking, C: confined to bed), self-care (A: no problems with self-care, B: some problems with washing and dressing, C: unable to wash or dress), usual activities (A: no problems performing usual activities, B: some problems performing usual activities, C: unable to perform usual activities). Mild frailty was defined with 1–2 “B,” moderate with 3 “B” and severe with ≥1 “C”; ^8^: The health state is based on the 3-level version of the EQ-5D (EQ-5d-3L) instrument assessing 5 dimensions (mobility, self-care, usual activities, pain/discomfort and anxiety/depression) ranging from 0 (worst) to 100 (best); ^9^: acute illness includes sepsis, peritonitis, surgery, etc.; ^10^: severity is defined with modified criteria (WCC > 15 and/or Cr > 133 and/or T > 38.5). CDI: toxigenic *C. difficile* infection, GDH: non-toxigenic *C. difficile* infection, IBD: inflammatory bowel disease, DC: diarrhea controls, HC: healthy controls, IQR: interquartile range, BMI: body mass index, CVD: cardiovascular disease, CKD: chronic kidney disease, GI: gastro-intestinal, NA: not applicable, BP: blood pressure, WCC: white cell count, CRP: C-reactive protein, PLT: platelet, PPI: proton-pump inhibitor.

**Table 2 ijms-25-12653-t002:** Functions of the CDI unique 45 gene set. Most genes were downregulated in CDI compared with all control groups (GDH, IBD, DC, and HC), and only three genes (*ARLNC1*, *NSUN7,* and *OLFM4*) were upregulated.

Gene Symbol	Category, Subcategory	Function	Reference
*ACTL6A/BAF53A*	Genomic stability, T-cell memory	Component of Brahma-associated factor (BAF) complexes, which participates in chromatin accessibility for pluripotency transcription factors (e.g., NANOG) by ATP-dependent nucleosome eviction. For instance, BAF regulates T cell differentiation to effector subsets and establishment of long-lived tissue memory T cells and maintains pluripotency by revoking differentiation of acute promyelocytic leukemia cells. Oncogenic activity in many cancers.	[52,53,54,55]
*ALDH9A1*	Metabolism—FAO, Redox	Least studied aldehyde dehydrogenase involved in carnitine biosynthesis, a fatty acid transporter to the mitochondria (mt) and a potent antioxidant.	[56]
*ARLNC1*	Genomic stability	Non-coding RNA that stabilizes the androgen receptor RNA in prostate cancer.	[57]
*BLCAP*	Genomic stability	A tumor suppressor/apoptosis inducer which interacts with Rb1, a chromatin stabilizer, and STAT3, which mediates the transcription of many factors, including Bcl-2 family proteins, cyclins, and matrix metalloproteinases.	[58,59,60]
*CD19*	Humoral immunity	Enhances B-cell expansion and antibody secretion via interacting with B-cell receptor (BCR) and CD21 and activating PI3k/*Akt* signaling. Deficiency impairs humoral memory.	[61,62]
*CLDN5*	Cytoskeleton, Humoral immunity	Component of tight junctions in epithelial and endothelial cells. Downregulation in enterocytes during inflammatory colitis. Also expressed by lymphocytes and monocytes, but not granulocytes, co-localized with tight-junction scaffold proteins at the cell membrane, and mRNA and protein levels increase in relapse of multiple sclerosis.	[63,64,65]
*COQ6*	Metabolism—OXPHOS	Necessary for coenzyme Q biosynthesis, a potent antioxidant and lipid oxidation enzyme.	[66]
*COQ10A*	Metabolism—OXPHOS	Required for coenzyme Q function of the mt respiratory chain. Potential marker of sepsis-induced cardiomyopathy.	[67,68]
*CYB561A3*	Metabolism—OXPHOS, Mt homeostasis	Ferrireductase necessary for mt respiration. Knockdown causes iron starvation with subsequent lysosomal and mt damage. Downregulated in HPV-induced warts and sepsis. Highly expressed in Burkitt lymphoma and involved in B cell proliferation/differentiation via cellular iron homeostasis.	[69,70,71]
*DYNLT3*	Cytoskeleton, T-cell memory	Subunit of the dynein complex, which transports organelles and cargos along the microtubule. Dynein promotes lymphocyte polarization during the immune synapse formation and asymmetric CD8+ T cell division, generating memory cells. An age-related oncogene in breast cancer.	[72,73,74]
*EI24*	Mt homeostasis	Endoplasmic reticulum (ER) protein promoting ER-mt contact for excessive calcium transfer during DNA damage, resulting in p53-mediated mt-induced apoptosis. Links ubiquitin-proteasome system (UPS) with autophagy via degradation of RING E3 ligases, including RNF41.	[75,76]
*FANCF*	Genomic stability, Mt homeostasis	Subunit of the Fanconi anemia core complex (E3 ubiquitin ligase), which repairs DNA inter-strand crosslinks. Mutations have been associated with cancer. Also factor of selective autophagy (mitophagy)	[77,78,79]
*FN3KRP*	Metabolism	Reverts non-enzymatic glycation of proteins, restoring their function. Increased expression has a potential protective role, promoting longevity and reducing pulmonary inflammation in smokers.	[80,81]
*GPA33*	T-cell memory	Expressed mainly by CD4+ T cells and associated with a central memory phenotype. TCR activation potentially reduces expression and marks an effector phenotype, which is required for effective humoral immunity.	[82,83]
*GPAA1*	Metabolism—FAO (inferred), Humoral immunity	Subunit of the GPIT complex, which anchors proteins lacking a transmembrane domain to the cell membrane. Heparin sulfate (HS), a glycosaminoglycan (sGAG) linked to a core protein (e.g., proteoglycan) regulating a plethora of cell functions, including lipid metabolism and autophagy, is GPI-anchored. HS strengthens IL-21 signaling in B cells, promoting differentiation towards antibody-secreting cells in germinal centers.	[51,84,85]
*LDLRAP1*	Metabolism—FAO (inferred)	Interacts with the cytoplasmic tail of the low-density lipoprotein receptor (LDLR) and drives LDL-LDLR endocytosis in polarized cells such as lymphocytes.	[86]
*LPAR5*	Humoral immunity	Negatively regulates TCR signaling and effector functions in CD8 by modifying the cytoskeleton and reprogramming metabolism. Also impairs BCR-induced calcium mobilization and antigen-dependent antibody production in B cells.	[87,88,89]
*MOCS1*	Redox	Subunit of the molybdenum co-factor which is a redox-active prosthetic in the active site of many enzymes with vital metabolic roles, including purine and sulfur amino acids catabolism in humans and anaerobic respiration in bacteria.	[90,91,92]
*MRNIP*	Genomic stability	Concentrates the MRN complex on sites of DNA damage to repair double-strand breaks.	[93]
*NAA30*	Mt homeostasis	Catalytic subunit of the NatC complex, which mediates N-terminal acetylation of a big repertoire of substrates, including mitochondrial proteins, preventing their degradation.	[94,95]
*NDUFAF3*	Metabolism—OXPHOS	Participates in the assembly of Complex I of mitochondrial oxidative phosphorylation.	[96]
*NSUN7*	Genomic stability/Metabolism—glycolysis	A methyltransferase of RNA acting during metabolic stress to stabilize regulatory and other types of RNA and transcriptional co-activators for rapid responses to the need for energy (e.g., upregulation of glycolysis-related enzymes).	[97]
*OLFM4*	T-cell memory (inferred)	Glycoprotein which is strongly expressed in the GI tract and prostate. mRNA and protein increased in neutrophils of mice with *C. difficile* infection, promoting pathogenesis and increasing morbidity and mortality. *OLFM4* was higher in serum of patients with *C. difficile* infection. *OLFM4* downregulates the Wnt pathway in gastrointestinal malignancies. Wnt/β-catenin signals the generation of memory stem cells in CD8 subsets.	[98,99,100]
*PEX3*	Metabolism—FAO	Required for the biogenesis of the peroxisome, where very long fatty acids are oxidized and transfer proteins to organelle membranes.	[101]
*PHF5A*	Humoral immunity, Genomic stability	Component of the spliceosome that regulates DNA repair during antibody class switch recombination and essential for B lymphopoiesis and Ig heavy chain production. In pancreatic cancer stem cells, the PAF1-PHF5A-DDX3 complex regulates the expression of self-renewal genes, including β-catenin.	[102,103,104]
*PIGU*	Metabolism, Humoral immunity	Subunit of the GPIT complex (see *GPAA1*).	
*PMPCA*	Mt homeostasis	Subunit of the mitochondrial processing peptidase (MPP), which cleaves almost all proteins transported from the cytosol into mitochondria to a fully mature state.	[105]
*PRICKLE1*	Cytoskeleton	A planar polarity gene which causes cytoskeleton restructuring through non-canonical Wnt signaling.	[106]
*RHOBTB2/DBC2*	Metabolism	Atypical Rho-like GTPase and substrate adaptor (BTB domain) to cullin 3 (scaffold) and RBX1/ROC1 (RING finger protein/catalytic domain of the E3 ubiquitin ligase) of the CRL3 complex, which mediates the ubiquitination and degradation of many proteins.	[107,108]
*RNF41/NRDP1*	Mt homeostasis	A RING-finger E3 ligase that promotes mitophagy via complex formation with CLEC16A and USP8. Involved in early termination of TCR signaling in CD8 cells, reducing IL-2 and IFN-γ production and TLR-mediated production of type I interferon (α/β).	[109,110,111]
*RNFT2/TMEM118*	Metabolism	Poorly characterized RING finger E3 ligase, which promotes degradation of IL-3Rα protecting from the deleterious effects of IL-3.	[112]
*RUSF1/C16orf58*	Not known	Poorly characterized protein. Interacts with *TMEM9* in the BioGRID4.4 database.	
*SLC2A11*	Metabolism—glycolysis (inferred)	Transports glucose and fructose. Limited literature studies.	[113]
*SMIM20/MITRAC7*	Metabolism—OXPHOS	Stabilizes COX1 in cytochrome c of the electron transport chain. All three components of cytochrome c (COX1/2/3) are encoded in mitochondria.	[114]
*SPIB*	Humoral immunity	Transcription factor regulating genes involved in the generation of memory B cells and maintenance of humoral memory (upregulates anti-apoptosis and autophagy genes). Transcription is suppressed by the Wnt pathway/SIX1 in Hodgkin lymphoma. Required for TLR7/9 mediated type I interferon (IFN-I) production in plasmacytoid dendritic cells.	[115,116,117]
*STING1/STING*	T-cell memory, Humoral immunity	Detects cytosolic nucleic acids and activates IFN-I production to clear viral infections. Herpes simplex virus 1 (HSV-1) inhibits β-catenin of the canonical Wnt pathway, which activates cGAS/STING signaling. STING signaling promotes antibody production through BCR regulation and T cell memory depending on TCR signal strength.	[118,119]
*TBCB*	Cytoskeleton	A tubulin chaperon that interacts with TBCE to degrade α-tubulin independent of energy.	[120]
*TMEM9*	T-cell memory	Transmembrane (organelles and cell) protein that activates the canonical Wnt pathway to promote intestinal and liver tumorigenesis and liver regeneration.	[121,122,123]
*TOP1MT*	Mt homeostasis	Mitochondrial (exclusively) topoisomerase that relaxes mtDNA supercoiling. Reduced expression results in the release of mtDNA to the cytosol, which activates the cGAS/STING pathway. Genetic variants have a significant effect on the expression of proteins involved in oxidative phosphorylation with irreversible impairment of mitochondrial respiration. Overexpressed in cancer.	[124,125,126,127]
*TSPAN31*	Cytoskeleton (inferred)	A poorly studied tetraspanin activating transcription of oncogenic factors (*Akt*) in gastric cancer. Tetraspanins organize proteins in the cell membrane.	[128]
*TSR3*	Metabolism	An aminocarboxypropyl (acp) transferase catalyzes the last step in the biosynthesis of 1-methyl-3-(3-amino-3-carboxypropyl)-pseudouridine of the 18S rRNA.	[129]
*USP11*	Genomic stability	A deubiquitinase (ubiquitin-specific protease or USP) with a dual role in cancer which is irreplaceably involved in many important cellular functions, including DNA repair (e.g., stabilizing *BRCA2*), cell division (stabilizing RanBMP) and NF-κB signaling downregulation. Also facilitates differentiation to regulatory T cells.	[130,131]
*VHL*	Metabolism-glycolysis, T-cell memory	Component of a ubiquitination complex involved in the degradation of hypoxia-inducible factor (HIF), which regulates gene expression and upregulates glycolytic metabolism during low oxygen conditions in memory CD8 cells. In renal cell carcinoma, it promotes STING degradation, while HIF increases due to loss of VHL, results in mtDNA leak and subsequent cGAS-STING activation.	[132,133,134]
*ZNF252P*	Not known	Pseudogene with unknown function and no relevant literature studies.	
*ZNF684*	Not known	Interacts with amyloid-β, TRIM28, and another 14 interactors in the BioGRID database. No literature studies on its function.	[135,136]

FAO: fatty acid oxidation, redox: reduction-oxidation, OXPHOS: oxidative phosphorylation, CDI: toxigenic *C. difficile* infection, GDH: non-toxigenic *C. difficile* infection, IBD: inflammatory bowel disease, DC: diarrhea controls, HC: healthy controls.

## Data Availability

Microarray data are available on Gene Expression Omnibus GEO, accession GSE276395, using the access token ‘ihixiuiajbgplar’ without quotes.

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
