# Peer review of "Gene Expression Dysregulation in Whole Blood of Patients with *Clostridioides difficile* Infection"

_ijms, 2024, doi:10.3390/ijms252312653_

Round 1

Reviewer 1 Report

Comments and Suggestions for Authors

The manuscript includes an extensive analysis of gene expression in the blood of patients with Clostridioides difficile infection. The study included analysis of samples and data obtained from 200 patients. Together with the analysis of gene expression, extensive literature and computer analyses of selected genes and metabolic pathways were conducted. As the Authors themselves pointed out, the study group was heterogeneous; however, the detailed description of the analyzed groups allowed for the observation of significant correlations. The publication has a very comprehensive bibliography, comprising 231 references, but the citations used seem well-justified. This is a well-written manuscript, presented in a clear and comprehensible language.

I have only a few minor editorial suggestions:
- Figure 2: Due to its large size, it is difficult to analyze. It might be worth considering splitting it into two separate figures.
- Figure 3: In the final version of the publication, the Venn diagram in panel A could be slightly smaller.
- Figures 4 and 6: The Venn diagrams could also be slightly smaller in the final version of the manuscript.
- Figure S7 (Supplementary Materials): The charts (left colunm) are of low resolution and difficult to read.

Author Response

Comment: I have only a few minor editorial suggestions:
- Figure 2: Due to its large size, it is difficult to analyze. It might be worth considering splitting it into two separate figures.
- Figure 3: In the final version of the publication, the Venn diagram in panel A could be slightly smaller.
- Figures 4 and 6: The Venn diagrams could also be slightly smaller in the final version of the manuscript.
- Figure S7 (Supplementary Materials): The charts (left colunm) are of low resolution and difficult to read.

Response: Thank you very much for your review and input.
- We split Figure 2 in to two separate figures (Figures 3 and 4 in current v2),
- decreased the size of the Venn diagrams (Figures 5-a, 6 and 8-a in v2)
- increased the size of Figure S7 (Figure S6 in v2).
The figures are clearer and easier to read. We appreciate that your suggestions have significantly improved our paper.

Reviewer 2 Report

Comments and Suggestions for Authors

The authors have used samples stool and blood collected in several years at the UK hospitals. The groups patients based on the non-diarrhoea or diarrhoea-origin. They compared the transcriptomes based on microarray assays and performed gene-ser enrichment analysis. Moreover, they did a literature research to described the most interesting genes.

The study involves a large number of samples and may comparisons. However, non of the results have been confirmed and due to very mild threshold on the significance and fold change, this study has a very weak support for the authors' claims.

Please, see my comments below.

#1. The PCA is CRUCIAL for this study and should be presented as a main figure. It shows, that the samples DO NOT cluster according to the group and little change is expected performing the comparisons. Therefore the FDR threshold should be more stringent, as well as log2|FC|.

#2. Comparing the diarrhoea groups to non-diarrhoea revealed neutrophil-related pathways - which, in my opinion, is expected for patients with inflammation or current infection. The neutrophil count should be also included in Table 1 and maybe taken into account in the expression analysis.

#3 The adjusted p.value threshold of 0.05 and log2|FC| of 0.5 is way too relaxed for such study. How the results would look like if you be more stringent with FDR < 0.01 and log2|FC| >=1? There will be only few genes left in most cases... Otherwise at least the 12 genes should be validated with qPCR.

#4 The volcano plots presented as main figure look like they are showing un-adjusted p-value, and for sure the threshold of 0.05 significants does not match the coloring, as -log10(0.05) = 1.3 whereas the pllots show the threshold around 2, which would fit to 0.01 p-value (or FDR?) threshold.

#5 The IPA figures are impossible to read due to small font size

#6  I wanted to check the results but the "RMAExpressionSet.RDS" seems corrupted. Please, check all the files.

Minor remarks

- shouldn't Latin be written in italics?

- gene names SHOULD be written in italics

Author Response

Comment 1: The PCA is CRUCIAL for this study and should be presented as a main figure. It shows, that the samples DO NOT cluster according to the group and little change is expected performing the comparisons. Therefore the FDR threshold should be more stringent, as well as log2|FC|.

Response 1: Thank you for the comment. The PCA has been moved within the main text (Figure 2).

We agree that PCA shows significant overlap of groups and we have pointed this out in the first sentence of our results describing gene expression findings (paragraph 2.2). We expected this overlap given that samples were from peripheral blood while the main site of inflammation was the gut. We also knew that the current study design would not identify unique genes but probably pathways. We appreciate that the FDR threshold and log2|FC| are arbitrary and there has been a lot of debate around appropriate cutoffs. We tend to use more stringent thresholds if we attempt to find biomarker genes. However, we also appreciate that minor changes in gene expression may have significant effects on cellular functions and as long as identification of mechanisms was the aim of this work, we decided that fold-changes should be kept as low as possible. As to the FDR, all genes in the presented sets have adjusted p value much less than 0.01 as shown in the Volcano plots.

We understand that different groups would have treated the data differently and we are looking forward to seeing their approach. The tables of statistics for all genes from the differential expression analyses are available in the GitHub repository folder ‘/output/tables’, where interested researchers are free to filter based on whichever log2|FC| and adjusted p-value thresholds they consider most appropriate.

Comment 2: Comparing the diarrhoea groups to non-diarrhoea revealed neutrophil-related pathways - which, in my opinion, is expected for patients with inflammation or current infection. The neutrophil count should be also included in Table 1 and maybe taken into account in the expression analysis.

Response 2: Indeed, we were also not surprised to see overexpression of neutrophil-related pathways in patients with diarrhoea compared to healthy controls. As you very well pointed out this is expected for patients with inflammation or infection and we have presented the evidence in our mini-review.

All clinical parameters were taken from patients’ clinical records and neutrophil count was not collected. However, white cell count is a good reflection of neutrophil count in this group of patients.

Comment 3: The adjusted p.value threshold of 0.05 and log2|FC| of 0.5 is way too relaxed for such study. How the results would look like if you be more stringent with FDR < 0.01 and log2|FC| >=1? There will be only few genes left in most cases... Otherwise at least the 12 genes should be validated with qPCR.

Response 3: Please see response 1 regarding our argument about the choice of more relaxed thresholds. Unfortunately we do not have any samples left to perform qPCR as we reported in our study limitations. We added adjusted p-values and log2FC for comparisons between diarrhoea groups and healthy controls in Table S6 to demonstrate that these genes satisfy more stringent filtering criteria.

Comment 4: The volcano plots presented as main figure look like they are showing un-adjusted p-value, and for sure the threshold of 0.05 significants does not match the coloring, as -log10(0.05) = 1.3 whereas the pllots show the threshold around 2, which would fit to 0.01 p-value (or FDR?) threshold.

Response 4: Yes, this is an astute observation. The volcano plots contain unadjusted p-values, as information can be lost by FDR-adjustment truncating some p-values to the same value. The colouring is nonetheless based on FDR-adjusted p-values, as is appropriate when conducting multiple hypothesis tests.

This may be verified by examining the code for the custom ‘volcano()’ R function in the script ‘03_manuscript_figures.R’, where points are coloured if they satisfy the ‘pval_under_05 == TRUE’ logical criterion, which is defined in the script ‘02_pca_dea.R’  using the FDR-adjusted p-values.

Comment 5: The IPA figures are impossible to read due to small font size

Response 5: Thank you for noticing. The IPA figures have been re-sized.

Comment 6: I wanted to check the results but the "RMAExpressionSet.RDS" seems corrupted. Please, check all the files.

Response 6:  Unfortunately we are not able to reproduce this problem, which is regrettable as we very much wish the analysis to be reproducible. To test this from scratch, we cloned the GitHub repository on the 8th November 2024, updated R to version 4.2.2 and updated all packages listed in the ‘install’ folder. We proceeded to run the script ‘02_pca_dea.R’, which includes importing the file via the R function readRDS("processed/RMAExpressionSet.RDS"), and we did not encounter any issues with corruption, or any error messages.

It may help to inspect the R package versions we used for this analysis, which can be found in the file ‘output/sessionInfo/02_pca_dea_sessionInfo.txt’.

Another possible solution would be to download the CEL files from GEO accession GSE276395 (access token ‘ihixiuiajbgplar’), save them in a  folder ‘raw/GSE276395_RAW/’ and recreate ‘RMAExpressionSet.RDS’ using the script ‘01_process_cel.R’. We appreciate that this would be laborious and potentially time-consuming, however.

To simplify this step, once the manuscript has been accepted for publication and we can lift the embargo on the GEO repository, we will update the code to automate downloading and processing of the CEL files using the ‘GEOquery’ R package.

Comment 7: Minor remarks

- shouldn't Latin be written in italics?

- gene names SHOULD be written in italics

Response 7: Thank you for the minor remarks. We have updated the revised version with italics for Latin and gene names.

Reviewer 3 Report

Comments and Suggestions for Authors

Very well written and organized paper.

Comments:

1. Figures 2(e-h), Figure 3 and 5 (d-f) and S7 are not readable when printed and blurry under magnification on the computer screen, increase fonts and resolution.

2. Lines 437- 442. Were all RNA samples processed together or in batches? If in batches, were all experimental groups evenly represented in each batch, and how batch effects were evaluated and corrected? Please add a paragraph addressing the batching design and batch correction.

3. Since the figures and tables are large I would recommend to reformat the paper to make it easier to read. Font sizes need to be increased in the figures (see comment1), but can be significantly decreased in the tables. Please make the main text body easily distinguishable from figure and table legends by using different font size and skipping the lines between the legends and the main text, but not within the legends (example of poor formatting lines 153-182).

Author Response

Comment 1: Figures 2(e-h), Figure 3 and 5 (d-f) and S7 are not readable when printed and blurry under magnification on the computer screen, increase fonts and resolution.

Response 1: Thank you for your comment. We have amended Figures 2e-h (Figure 4 in current V2), Figure 3 (Figure 5 in V2) and 5d-f (Figure 7d-f in V2) and S7 (Figure S6 in V2) and they are easier to read.

Comment 2: Lines 437- 442. Were all RNA samples processed together or in batches? If in batches, were all experimental groups evenly represented in each batch, and how batch effects were evaluated and corrected? Please add a paragraph addressing the batching design and batch correction.

Response 2: This is an important question. All RNA samples were processed in batches of 16-20 samples within a period of a month (January 2017).

Microarrays were scanned across 8 batches over a timespan of 42 days. Unfortunately not all experimental groups were represented across all microarray batches. The contingency table for the number of samples per microarray batch by experimental condition is as follows:

CDI

GDH

IBD

DC

HC

Batch 1

25

0

0

0

23

Batch 2

12

0

0

7

0

Batch 3

11

0

0

7

0

Batch 4

6

8

31

0

12

Batch 5

0

29

9

0

10

Batch 6

6

0

0

11

6

Batch 7

11

0

0

4

0

Batch 8

7

0

0

16

0

A design matrix containing both experimental group and batch ID variables is not of full rank, though it is still possible to adjust for batch in the comparisons made in this study. To facilitate further exploration of this, we have added the batch date metadata to the GitHub repository in the file ‘raw/metadata_batch.csv’.

We originally assessed potential effects of batch by principal component analysis, where no clear association was found in pairwise PC plots for all principal components up to PC9. We have added text clarifying this in lines 480-482 and added supplementary Figure S7, with accompanying code in the GitHub repository.

Comment 3: Since the figures and tables are large I would recommend to reformat the paper to make it easier to read. Font sizes need to be increased in the figures (see comment1), but can be significantly decreased in the tables. Please make the main text body easily distinguishable from figure and table legends by using different font size and skipping the lines between the legends and the main text, but not within the legends (example of poor formatting lines 153-182).

Response 3:  We agree that the paper was not easy to read at its previous version. We tried to follow the MDPI template but probably some styles were altered during copy-paste. We really appreciate your suggestions and we have reformatted the document accordingly.